# Beneficial Effects of Three Dietary Cyclodextrins on Preventing Fat Accumulation and Remodeling Gut Microbiota in Mice Fed a High-Fat Diet

**DOI:** 10.3390/foods11081118

**Published:** 2022-04-13

**Authors:** Tong Zhu, Baixi Zhang, Yan Feng, Zhaofeng Li, Xiaoshu Tang, Xiaofeng Ban, Haocun Kong, Caiming Li

**Affiliations:** 1School of Food Science and Technology, Jiangnan University, Wuxi 214122, China; 15388545375@163.com (T.Z.); zbx@jiangnan.edu.cn (B.Z.); fengyanjiangnan@163.com (Y.F.); zfli@jiangnan.edu.cn (Z.L.); txs@jiangnan.edu.cn (X.T.); banxiaofeng521@163.com (X.B.); haocunkong@jiangnan.edu.cn (H.K.); 2Key Laboratory of Synergetic and Biological Colloids, Ministry of Education, Wuxi 214122, China; 3Collaborative Innovation Center of Food Safety and Quality Control, Jiangnan University, Wuxi 214122, China; 4National Engineering Research Center for Functional Food, Jiangnan University, Wuxi 214122, China

**Keywords:** cyclodextrin, antiobesity, prebiotic, microbiota

## Abstract

Globally, obesity and its metabolic complications, which are intimately linked to diet, are major public health problems. Cyclodextrins (CDs) are cyclic oligosaccharides consisting of (α-1,4)-linked D-glucopyranose units that can reduce fat bioavailability and affect metabolism by improving intestinal flora as prebiotics. We compared the effects of three CDs on preventing fat accumulation and remodeling gut microbiota in a high-fat diet-fed C57BL/6J mouse model. α-CD maximized energy expenditure by 12.53%, caused the RER value to drop from 0.814 to 0.788, and increased the proportion of fatty acid oxidation for energy supply. β-CD supplementation resulted in a marked 24.53% reduction in weight gain and a decrease in epididymal-fat-relative weight from 3.76% to 2.09%. It also minimized ectopic fat deposition and improved blood lipid parameters. γ-CD maximized the concentration of SCFAs in the cecum from 6.29 to 15.31 μmol/g. All three CDs reduced the abundance ratio of Firmicutes and Bacteroidetes to a low-fat diet level, increased the abundance of *Lactobacillus* and *Akkermansia*, and reduced the abundance of *Allobaculum* and *Ruminococcus*. These findings imply that a combination of multiple CDs may exert superior effects as a potential strategy for obesity prevention.

## 1. Introduction

Obesity, a complex multifactorial disease, is characterized by an abnormally high percentage of body weight, which is attributed to the size and increased number of fat cells in the body. It is often associated with type 2 diabetes, hyperlipidemia, cardiovascular disease, and cancer [1,2,3]. Globally, obesity and its metabolic complications are a major public health concern [4]. In 2016, more than 1.9 billion adults (older than 18 years of age) were found to be overweight. Among them, more than 650 million were obese. Worldwide obesity has nearly tripled since 1975, and the number continues to grow [5]. Obesity prevention and management requires the consumption of a healthy and energy-balanced diet, as well as adequate physical activity levels. A major impediment to the management of obesity is excess caloric consumption, especially due to the intake of high-fat foods [6].

Cyclodextrins (CDs) are cyclic oligosaccharides consisting of (α-1,4)-linked D-glucopyranose units with slightly conical hydrophobic cavity structures. The most common natural CDs are α-CD, β-CD, and γ-CD, which consist of six, seven, and eight D-glucopyranose units [7]. CDs are non-toxic, odorless, and tasteless and are widely used as food additives to improve sensory, nutritional, and performance properties, for example, by reducing foam, stabilizing emulsions, and lessening enzymatic browning [8,9,10,11]. The cyclodextrins α-CD and β-CD are most likely resistant to digestion by salivary or pancreatic amylase; moreover, they have a fairly low absorption rate in the intestines. Thus, they can enter the large intestine intact and be hydrolyzed by enzymes produced by intestinal flora and metabolized into short-chain fatty acids (SCFAs). [12,13]. In addition, due to their special structures, they can form inclusion complexes with many hydrophobic guest molecules, such as cholesterol and fatty acids [7,14,15,16,17,18]. The effect of CDs on gut microbes and dietary fat encapsulation gives them potential in the prevention and treatment of obesity. α-CD has been used as a dietary fiber for weight control and is marketed in the US, Canada, and Australia [19]. The high affinity of α-CD for fatty acids can reduce the bioavailability of dietary lipids [6,20,21], which is beneficial for obese people and populations at high risk of cardiovascular diseases. β-CD has a role in regulating lipid levels, but its use as nutraceutical is limited by possible hepatotoxic effects [22]. Although the luminal and/or epithelial enzymes of the gastrointestinal tract are capable of digesting γ-CD into glucose, some γ-CD or the hydrolyzed fraction is still able to reach the colon [23,24]. Intestinal microbiota are able to ferment indigested components, thus causing changes in the bacterial population [25]. Therefore, based on the application of α-CD in weight control and the similar structure of α-CD, β-CD, and γ-CD, it would be valuable to explore the efficacy of CDs in the prevention and treatment of obesity.

The gut microbiota directly mediates the regulation of enriched physiological processes through interactions with the host and/or indirectly through metabolites, such as short-chain fatty acids (SCFAs). Alterations in gut microbiota structure and its functional abnormalities can lead to impaired energy homeostasis in the host, thereby contributing to metabolic disease promotion [26,27,28,29]. Changes in intestinal microbiota, such as an increase in the Firmicute/Bacteroidetes ratio, play an important role in the pathophysiology of obesity [30]. The increased conversion of energy and lipogenesis, as well as the increased intestinal permeability, followed by inflammatory responses and hormonal changes, provoked endotoxemia, and reduced SCFAs levels, can all explain the accelerated course of obesity [31,32].

Based on the encapsulation effect of CDs on dietary fat and their ability to be fermented by intestinal flora, we suggest that dietary supplementation with CDs has potential in preventing fat accumulation and remodeling gut microbiota in mice on a high-fat diet. In this study, we investigated the effects of three CDs on fat accumulation, energy expenditure, SCFA levels in cecum contents, lipid metabolism-related gene expression, and gut microbiota abundance in a high-fat diet-fed C57BL/6J mouse model. Our findings will inform effective strategies for obesity prevention.

## 2. Materials and Methods

### 2.1. Experimental Animals

A total of 36 6-week-old male C57BL/6JNifdc mice were purchased from Vitalriver Co., Ltd., (Beijing, China). They were kept in a specific pathogen-free (SPF) environment at the Jiangnan University Laboratory Animal Center (license no. SYXK(Su)2021-0056, Wuxi, China) at a constant temperature (22−26 ℃) and humidity (40−60%). The Laboratory Animal Ethics Committee of Jiangnan University approved this study (license number JN.No 20201030c0480201) (295).

### 2.2. Procedures

After 1 week of acclimatization, the 36 mice were randomly assigned into 6 groups: (i) control (low-fat diet), (ii) HFD baseline (45% high-fat diet without cellulose), (iii) fiber (45% high-fat diet), (iv) ACD (45% high-fat diet that replaced cellulose with α-CD), (v) BCD (45% high-fat diet that replaced cellulose with β-CD), and (vi) YCD (45% high-fat diet that replaced cellulose with γ-CD). All mice diets were produced by Trophic Animal Feed High-Tech Co., Ltd., Nantong, China. Each group was fed its respective diet and watered ad libitum during the 9 weeks of the experiment. The fasting weights of mice were measured at the same time every week. Metabolic rate monitoring was performed 5 days before the end of the experiment. Feces of mice in each group were obtained in nuclease-free tubes one day before the end of the experiment and were stored at −80 °C after liquid nitrogen freezing. Mice were fasted for 12–14 h, anesthetized by intraperitoneal injection of pentobarbital sodium (80 mg/kg body weight), sacrificed, and dissected for sampling at the end of the experiment. Eyeball exsanguination was used to extract retro-orbital blood samples right away. To obtain serum, clotted blood samples were centrifuged (3500 g) at 4 °C for 10 min. The liver, brown adipose tissue, epididymal adipose tissue, and cecum and its contents were then carefully removed and either frozen in liquid nitrogen or preserved in 4% paraformaldehyde for subsequent investigation.

### 2.3. Metabolic Rate Monitoring

A comprehensive lab animal monitoring system (CLAMS, Columbus Instruments Inc., Columbus, OH, USA) was used to track the metabolic rates of the mice. Each mouse was housed in its own CLAMS chamber with free access to water and a customized diet. Officially, monitoring started after 12 h of adaptation and lasted for 48 h. To quantify energy expenditure and the respiratory exchange ratio (RER), each chamber’s oxygen consumption and carbon dioxide production in expired air were measured.

### 2.4. Biochemical Analysis in Serum and Liver

Serum levels of triglycerides (TG), total cholesterol (TC), low-density lipoprotein cholesterol (LDL-C), high-density lipoprotein cholesterol (HDL-C), alanine aminotransferase (ALT), and aspartate aminotransferase (AST) were determined by a BK-400 automatic biochemical analyzer.

Livers were homogenized at 4 °C for further analysis. Free fatty acid (FFA), TG, TC, and hepatic glycogen content was quantified using corresponding commercial kits (Jiancheng Bioengineering Institute, Nanjing, China).

### 2.5. Histopathological Analysis

All tissues were fixed in 4% paraformaldehyde for more than 24 h. Gradually, the brown adipose and epididymal adipose tissues were dehydrated, embedded in paraffin, and sliced into thin sections that were stained with H&E. Liver tissues were frozen, sectioned, dried, and protected from light for oil red staining and, later, for hematoxylin staining.

### 2.6. Short-Chain Fatty Acid (SCFA) Analysis in Cecal Contents

Cecal contents (0.09 g) were suspended in 0.5 mL of pre-chilled saturated NaCl solution at 4 °C and left to stand in an ice bath for 30 min. Then, cecal content suspensions were homogenized, after which 10 μL of pre-chilled 10% (*v*/*v*) sulfuric acid solution was added at 4 °C. The acidified sample was extracted using 1 mL of anhydrous ether containing 1 mmol/L of internal standard (2-ethylbutyric acid), vortexed to mix for 30 s, and centrifuged at 4 °C for 15 min at 18,407× *g*. Then, 0.25 g of anhydrous sodium sulfate was added to remove water. Centrifugation was carried out at 18,407 g for 15 min at 4 °C, after which the supernatant was analyzed by gas chromatography for evaluation of short-chain fatty acids (acetate, propionate, isobutyrate, butyrate, isovalerate, and valerate).

### 2.7. S Ribosomal RNA Gene Sequencing

APCR reaction system was configured using 30 ng of qualified genomic DNA samples and the corresponding fusion primers: 341F (5′-ACTCCTACGGGAGGCAGCAG-3′) and 806R (5′-GGACTACHVGGGTWTCTAAT-3′). The PCR amplification products were purified and dissolved by Agencourt AMPure XP (Beckman Coulter™, Brea, CA, USA) magnetic beads and labeled to complete library construction. Libraries were tested for fragment range and concentration using an Agilent 2100 bioanalyzer (Agilent, Santa Clara, CA, USA). Libraries that passed the test were selected for sequencing on a HiSeq platform (BGI Genomics Co., Ltd., Shenzhen, China) according to insert sizes. Data were filtered, and the remaining high-quality clean data were used in post-analyses. Sequence splicing was performed using FLASH software (Fast Length Adjustment of Short reads, v1.2.11), which assembles pairs of reads from double-end sequences into a single sequence using overlap relationships to obtain tags of high-variation regions. Amplicon sequence variants (ASVs) were obtained, compared with the database, and annotated with the species using the DADA2 (divisive amplicon denoising algorithm) method in QIIME2. Based on OTU and annotation results, sample species complexity analysis, intergroup species variation analysis, association analysis, and functional prediction were performed. The free online platform of BGI offered the bioinformatics analysis methods.

### 2.8. Quantitative RT-PCR Analysis

Quantitative RT-PCR analysis was carried out with the corresponding commercial kits from Nanjing Vazyme Biotech Co.,Ltd. (Nanjing, China). Total RNA was extracted from the liver, and 2 μg of RNA was reverse-transcribed into cDNA. Quantitative RT-PCR was performed on a CFX96 Touch real-time PCR system using SYBR Green qPCR Master Mix. The expression level of each gene was normalized to the *β*-actin gene and calculated utilizing the 2^−ΔΔCt^ method. The specific forward and reverse primer sequences are shown in Table 1.

### 2.9. Statistical Analysis

Statistical analyses were performed with SPSS 22.0 software (SPSS Inc., Chicago, IL, USA). Statistical differences among multiple treatments (≥3) were assessed by ANOVA using the Duncan procedures. *p* ≤ 0.05 was considered statistically significant.

## 3. Results

### 3.1. β-CD Supplementation Prevented Weight Gain and Fat Accumulation

The effects of CD supplementation on body weight and fat accumulation were determined by comparing the physical parameters and histological analysis of mice fed each diet: control (low-fat diet), HFD baseline (45% high-fat diet without cellulose), fiber (45% high-fat diet), ACD (45% high-fat diet that replaced cellulose with α-CD), BCD (45% high-fat diet that replaced cellulose with β-CD), and YCD (45% high-fat diet that replaced cellulose with γ-CD). Increased weight gain was noticed in HFD baseline group mice compared to control group mice (Figure 1a). To establish the effects of CDs on non-obese individuals on a high-fat diet, the experiment was concluded at week 9, when differences in body weights between the HFD baseline and control groups did not reach 20% and did not meet the criteria for determining the obesity model. The β-CD-supplemented diet suppressed excess weight gain in mice fed a high-fat diet. After 9 weeks of feeding, body weight gain in the BCD group was 24.53% lower compared with that of the HFD baseline group (*p* < 0.05), and this effect did not depend on energy intake (Figure 1b,c). Throughout the experiment, cellulose, α-CD, and γ-CD supplementation had little effect on body weight gain (Figure 1b).

CDs altered ectopic lipid deposition without affecting the relative weight of the liver (Figure 2a,b). Oil red staining of liver tissue sections revealed that in three CDs groups, especially the BCD group, macrovesicular and microvesicular steatosis and ectopic fat deposits in hepatocytes were markedly reduced compared to the HFD baseline group.

The relative weights of the brown adipose tissue (BAT) were significantly lower in mice in the HFD baseline group compared to those in the control group (Figure 2c), and the supplementation of cellulose and α-CD led to an increase in this index (*p* > 0.05). H&E-stained sections of BAT (Figure 2a) showed that the high-fat diet increased the size of lipid droplets in brown adipocytes, which squeezed the capillary network and nuclei, leading to nuclear displacement. α-CD, β-CD, and γ-CD supplementation attenuated this effect, in contrast to cellulose supplementation, which further increased the size of lipid droplets in brown adipocytes.

There were substantial differences in relative weights of epididymal fat between the BCD (2.09%) and HFD baseline (3.76%) groups (*p* < 0.01; Figure 2d). Histological analysis showed that sizes of adipocytes of epididymal adipose tissues were increased in the HFD baseline group; however, β-CD supplementation normalized this parameter (Figure 2a). Although cellulose and γ-CD supplementation reduced adipocyte diameters, their effects on relative weights of epididymal fats were not significant. Contrary to previous studies, we established that α-CD supplementation increased the relative weights of epididymal fat in mice.

The FFA levels in liver homogenates in the HFD baseline group was significantly lower than those in the control group, whereas α-CD supplementation restored them from 0.095 mmol/L to 0.137 mmol/L (Figure 3a). An increase in TG levels and a slight decrease in TC levels of the liver were observed in the HFD baseline group. CD supplementation resulted in an observable decrease in hepatic TG levels, and β-CD supplementation led to a decrease in TC levels (Figure 3b,c). Mice that received CDs and cellulose were also protected from excess liver glycogen accumulation (Figure 3d).

### 3.2. β-CD Supplementation Optimized Blood Parameters

Compared to the HFD baseline group, serum TC, TG, and HDL-C levels were significantly suppressed in the BCD group (*p* < 0.05), whereas serum LDL-C levels were reduced by 41.4% in the BCD group (*p* > 0.05) (Figure 4a–d). In addition, there were no significant differences in serum FFA levels between the groups (Figure 4g).

Although β-CD has widely been used as a food additive, it has been associated with hepatotoxic effects [22]. Therefore, we measured serum ALT and AST levels in this study (Figure 4e,f). There were no significant differences in serum ALT and AST levels compared with the control group; however, serum AST levels in the fiber group were significantly higher than those of the BCD group. Combined with histological analyses of the liver, these differences may be due to the accumulation of small lipid droplets in hepatocytes. These findings indicate that 6% β-CD supplementation does not cause damage to the hepatocytes.

### 3.3. α-CD Supplementation Promoted Energy Expenditure and Changed the Energy Supply Structure

Energy consumptions in each group of mice were monitored for more than 48 h (Figure 5d–e). Energy expenditure exhibited a clear circadian rhythm, with higher energy expenditure during the night than daytime. Energy expenditure increased with increasing fat content in the diet. α-CD supplementation enhanced energy expenditure in high-fat-diet mice, whereas γ-CD supplementation reduced energy expenditure to the level of the low-fat control group.

Then, respiratory exchange ratio (RER) was calculated to determine the energy supply ratio of carbohydrates to fatty acids (Figure 5a–c). Control group mice had RER values closer to 1 and preferred carbohydrate energy supply. As fat levels in the diet increased, the proportion of mice using fatty acids for oxidative energy supply increased, bringing the RER closer to 0.7. The RER of the fiber, ACD, and YCD groups were lower relative to those of the HFD baseline group, implying that the proportion of mice using fatty acids for oxidative energy supply had increased.

### 3.4. α-CD and γ-CD Supplementation Enhanced SCFA Levels in Cecum Contents

SCFAs in the cecum are primarily bacterial metabolites of undigested carbohydrate portions. These SCFAs exert a critical physiological effect on host health. The high-fat diet significantly decreased the concentrations of total SCFAs and acetate in cecum contents; however, it increased the concentrations of butyrate (*p* > 0.05) and valerate (*p* < 0.05) (Figure 6a,d,f,g). α-CD and γ-CD supplementation significantly increased the concentrations of total SCFAs in the cecum of mice fed a high-fat diet. The addition of 6% γ-CD increased SCFA levels by almost 1.44-fold, and the effect was significantly better than that of cellulose. This promotion effect was highly attributed to acetate and propionate concentrations (Figure 6a,b,g). α-CD supplementation significantly reduced isobutyrate concentrations but had no significant effects on butyrate concentrations (Figure 6d,e). β-CD supplementation had negligible effects on SCFA concentrations but only increased isovalerate concentrations (Figure 6e).

### 3.5. α-CD, β-CD, and γ-CD Supplementation Promotes Changes in Expression of Genes Involved in Lipid Metabolism

As shown in Figure 7, the relative expressions of *Srebp-1c* (sterol-regulatory element binding protein-1c), *Pparα* (peroxisome proliferator activated receptor-α), and *Pepck* (phosphoenolpyruvate carboxykinase) genes were determined to explore the effects of CD supplementation on lipid metabolism. SREBP-1C is a significant transcription factor known to regulate the expression of lipogenic enzymes, such as HMG-CoA, and correlation with intracellular cholesterol content. PPARα is a ligand-activated transcription factor that turns on the expression of genes related to fatty acid oxidation. PEPCK is directly involved in the synthesis of 3-phosphoglycerol and maintains triglyceride synthesis by FFA re-esterification and subsequent fat accumulation. Increased dietary lipid content did not affect *Srebp-1c* gene expression in the liver, but β-CD supplementation significantly upregulated it. CDs and cellulose supplementation introduced the *Pparα* and *Pepck* genes, which were upregulated by the high-fat diet back down to low.

### 3.6. α-CD, β-CD, and γ-CD Supplementation Restructured Fecal Microbiota Compositions

Fecal microbiota communities were investigated by 16S rRNA gene sequencing. The width and smoothness of the OTU rank curve reflect the richness and uniformity of gut microbes (Figure 8a). The high-fat diet without dietary cellulose reduced gut microbe richness and uniformity; however, cellulose supplementation reversed this effect. Contrary to γ-CD, the addition of α-CD and β-CD further weakened gut microbe richness.

Four indicators—Ace, Chao, Shannon, and Simpson—were calculated to describe alpha diversity within each group (Figure 8b–e). α-CD and β-CD supplementation were associated with a significant decrease in Ace and Chao indices, whereas cellulose and γ-CD supplementation increased these two indices. Compared to the HFD baseline group, Shannon indices were higher in the control, fiber, ACD, and YCD groups, especially in the fiber group, the value of which was even higher than that of the control group. The Shannon index value in the BCD group was decreased. The Simpson indices were expressively higher in the HFD baseline and BCD groups than in the control group, and the difference within the HFD baseline group was greater. The Simpson index fell back to a level comparable to that of the control group in the ACD group, with lower values in the YCD and fiber groups.

β diversity analysis is used to show differences in microbial community composition among different groups of samples. PCoA analysis showed that fecal microbiota in the control and fiber groups were clustered together and the ACD and BCD groups formed independent groups, whereas the YCD group was closer to the HFD baseline group (Figure 9b). The UPGMA cluster tree shows consistent results (Figure 9a).

Then, we analyzed microbial species composition for each group of samples. Analysis of differences in gut microbial composition at the phylum level (Figure 10a,b) revealed that the high-fat diet without dietary cellulose increased the relative abundance of actinobacteria and Firmicutes and decreasing the relative abundance of Bacteroidetes and Verrucomicrobia. Cellulose supplementation partially restored the relative abundance of Bacteroidetes; however, the relative abundance of Firmicutes increased from 81.49% to 87.14%. CD supplementation, especially β-CD, resulted in a remarkable increase in the relative abundance of Bacteroidetes and Verrucomicrobia, whereas the relative abundance of Firmicutes was decreased. Relative abundance ratios of Firmicutes and Bacteroidetes (F/B ratio) in the guts of each group exhibited a significantly higher F/B ratio in the HFD baseline group than in other groups, and cellulose and CD supplementation caused the F/B ratio to be adjusted downward to levels similar to those in the control group (Figure 10c).

Differences in gut microbial composition for each group were analyzed at the genus level (Figure 10d,e). CD supplementation significantly increased the relative abundance of Lactobacillus and Akkermansia in the gut, which is an effect that was not achieved by cellulose supplementation. CDs downregulated the relative abundance of Allobaculum and Ruminococcus compared to cellulose. Intakes of different types of CDs exerted different effects on intestinal microbe compositions. α-CD reduced the relative abundance of Bifidobacterium, whereas diets containing β-CD and γ-CD exerted the opposite effect. A high-fat diet containing α-CD enhanced the abundance of Lactobacillus, whereas β-CD suppressed the relative abundance of Allobaculum.

The statistical clustering method was used to cluster different samples with similar structures of dominant flora into one group for enterotype analysis. We clustered three kinds of enterotypes (Figure 10f), in which the control and fiber groups were clustered into one enterotype, the three CDs groups were clustered into one enterotype, and the HFD baseline group was clustered into one enterotype.

Linear discriminant analysis (LDA) effect size analysis (LEfSe) was used to distinguish biomarkers that were markedly different between groups (Figure 11a,b). Akkermansia, Coprococcus, and Butyricicoccus were identified as specific bacterial taxa of CDs.

According to the 16S rRNA gene sequences, metabolic functional pathways of the gut microbiota were doped-out. Compared to the HFD baseline group, we established that the relative abundance of genes related to the metabolism of cofactors and vitamins, metabolism of terpenoids and polyketides, transport and catabolism, and amino acid metabolism in the ACD, BCD, and YCD groups were significantly high. In addition, relative abundances of genes related to glycan biosynthesis and metabolism, as well as carbohydrate metabolism in BCD and YCD groups were notably different from those of the HFD baseline group (Figure 12).

## 4. Discussion

Globally, obesity and related metabolic disorders are growing health challenges. We compared the effects of three CDs on fat accumulation and gut microbiota in healthy, non-obese C57BL/6J mice fed a high-fat diet. Healthy, non-obese individuals on a high-fat diet represent a transitional group that is highly likely to transition to obesity or back to a healthier weight class through moderate diet and lifestyle changes. We supplemented C57BL/6J mice on a high-fat diet with 6% CDs, a dose equivalent to 35 g per day in humans, according to the Reagan-Shaw equations, which is a realistic dose in humans.

β-CD supplementation reduced excess weight gain due to the high-fat diet; moreover, it inhibited hepatic steatosis and fat accumulation in brown adipose tissues, as well as in epididymal fat tissue. This effect was not related to changes in energy intake and expenditure but may be attributed to lower metabolic efficiency. In addition, β-CD supplementation reduced serum TC, TG, LDL-C, and HDL-C levels. LDL-C is associated with incidences and extent of cardiovascular diseases and is considered to be the main causative factor for atherosclerosis [33]. The significance of TG as a risk factor for cardiovascular diseases has been extensively investigated in the general population. β-CD supplementation may result in a lower risk of cardiovascular disease. These beneficial effects may be provided by the high affinity of β-CD to fatty acids and cholesterol, which decreases the absorption of these lipids [15,23]. Cholesterol only forms complexes with β-CD because of a suitable steric fit in the cavity. Low levels of cholesterol in hepatocytes may upregulate the expression of the Srebp-1c gene, which could accelerate cholesterol synthesis. As a result of β-CD supplementation, the expression of the Pepck gene was inhibited, which may have attenuated gluconeogenesis and glyceroneogenesis in the liver. The former reduced hepatic glycogen accumulation, and the latter inhibited the synthesis of neutral fat and hepatic steatosis. Artiss et al. [34] showed that β-CD supplementation of a cholesterol-rich diet of rats reduced plasma triglyceride levels and promoted bile acid synthesis and excretion, as well as standardized biliary lipid secretion. β-CD exerted the same hypocholesterolemic effect with a low-fat diet [35]. Supplementation of feeds with 5% β-CD reduced cholesterol levels in the back fat, loin, belly, and ham portions of swine [36]. Garcia-Mediavilla et al., reported that supplementation of β-CD to a cholesterol-rich diet reduced TG levels and boosted bile acid synthesis, as well as excretion, and stabilized biliary lipid secretion. However, ALT and AST levels were remarkably increased and exhibited marked hepatotoxic effects [22]. In this study, we used serum ALT and AST to characterize the effects of β-CD on the liver and found that 6% β-CD supplementation does not cause damage to the hepatocytes. Histological analysis and relative weights of the liver did not reveal any hepatotoxic effects of β-CD. However during autopsy, gallbladder enlargement was detected in some mice in the BCD group, implying that β-CD as a nutraceutical requires more accurate dose control.

α-CD supplementation promoted energy expenditure in mice on a high-fat diet, increased the percentage of energy supplied by fat, and improved cecum total SCFA levels, which may explain its inhibitory effects on hepatic fat accumulation. BAT is a highly vascularized organ with plenteous mitochondria that make fuel oxidation primarily result in heat production instead of ATP generation [37]. Numerous spherical mitochondria surrounded by accessible lipid vacuoles and favorably vascularized brown adipocyte clusters support its thermogenic functions [38]. Obesity is bound-up with functional decay of BAT. Lipid droplet accumulation, capillary deficiency, and functional hypoxia in BAT can lead to its dysfunction [37]. Histologically, supplementation with 6% α-CD resulted in a more developed capillary network and smaller-diameter fat vacuoles in the BAT of mice fed a high-fat diet. This enhanced the susceptibility of the mitochondria to lipid droplets, which also had sufficient oxygen for thermogenesis, thereby delaying the disruption of thermogenic functions of BAT by the high-fat diet. In addition, downregulation of Pparα gene expression may have enhanced the expression of fatty acid oxidation-related genes, promoted β-oxidation of fatty acids in the liver, and increased energy expenditure. Higher levels of FFA in the liver may also imply enhanced fat mobilization.

We established that 6% α-CD supplementation had no significant effects on body weight gain, epididymal adipose tissue cell volume, and blood lipid levels in mice, which contrasts with earlier findings. Nihei et al. [13] reported that intake of 5.5% a-CD reduced weight gain and reduced epididymal adipose tissue weight in obese mice. Artiss et al. [34] found that compared to the high-fat group, after 6 weeks of feeding, the experimental group supplemented with a-CD had a 7.4% lower body weight and reduced plasma TG and TC levels, although there were no effects on food intake levels [34]. In low-density lipoprotein receptor (LDLR) knockout mice on a high-fat diet, marked reductions in plasma cholesterol (15.3%), free cholesterol l (20%), cholesterol esters (14%), and phospholipid (17.5%) levels were noticed in mice fed with α-CD compared to the control group [39]. A human study demonstrated the role of a-CD in weight control [6]. In a 2-month double-blind crossover study, 28 overweight but not obese compliant participants (8 males and 20 females) showed significant decreases in body weight (−0.4 ± 0.2 kg, *p* < 0.05) comparing active phase (tablets for oral administration of a-CD) to control phase (tablets for oral administration of cellulose). Differences in the effects of α-CD on weight control and blood lipid levels might be attributed to the choice of animal models, experimental period, differences in feed composition, and a-CD doses. C57BL/6J mice are highly sensitive to high-fat diets; therefore, induction of obesity is easier. Generally, their obesity begins with an increase in abdominal visceral fat deposition (centripetal obesity), followed by weight gain. This excess sensitivity might make it difficult for a-CD supplementation to inhibit excess weight gain as a result of a high-fat diet within a short period. To investigate the effects of CDs on fat accumulation in healthy, non-obese mice, the experiment was completed at 9 weeks, when the difference in fasting weight between the control and HFD baseline groups was 17.4%, before reaching obesity. In a previous study involving C57BL/6J mice [13], significant differences in body weight between the HFD group and HFD+ a-CD group did not appear until 14 weeks of feeding.

The beneficial effects of γ-CD on preventing fat accumulation may depend on the production of SCFAs. The high-fat diet significantly reduced total SCFA concentrations in cecum contents, particularly acetate and propionate, but elevated butyrate and valerate concentrations. Supplementation of the high-fat diet with 6% γ-CD increased total SCFA concentrations in cecum contents from 6.29 μmol/L to 15.31 μmol/L, whereas the concentrations of acetate and propionate were elevated by 485% and 241%, respectively. SCFAs are the major metabolites produced by specific intestinal flora fermenting dietary fibers, proteins, and peptides. They are a source of energy for certain host cells; can influence host lipid and blood glucose levels; and can also regulate the colonic environment, inflammation, metabolism, and immune functions, thereby mediating the regulatory roles of intestinal flora in host health [40,41]. Abnormalities in gut microbiota of obese populations lead to dysregulated short-chain fatty acid metabolism, thereby affecting various physiological functions that are associated with obesity [42,43]. Oral administration of acetate in obese and diabetic rats significantly reduced fat accumulation in adipose tissues, prevented fat accumulation in the liver, and improved glucose tolerance [44]. Findings from studies involving humans revealed that propionate reduces energy intake by exciting the release of PYY and GLP-1 from human colon cells, inhibiting weight gain, improving adipose tissue distribution in the abdomen, and reducing intrahepatic fat levels [45]. Furthermore, downregulation of Pparα and Pepck gene expression in the liver also attenuated hepatic steatosis

The gut microbiota affects host metabolism and contributes to the development of obesity [46]. The obese microbiome is more effective in obtaining energy from the diet [47]. Dietary factors have a direct impact on the composition of gut microbiota [48]. CDs may prevent obesity by optimizing intestinal flora. α-CD and β-CD are barely digested in the gastrointestinal tract but are fermented by the intestinal microflora, thus causing changes in the abundance of intestinal microbiota. Although γ-CD can be digested by luminal and/or epithelial enzymes of the gastrointestinal tract, it can influence carbohydrate digestion [49]. Additionally, γ-CD is incompletely digested in the upper gastrointestinal tract, and the indigested portions are fermented in the bowel, thereby altering bacterial populations [24].

The OTU rank curve shows that supplementation of α-CD with β-CD may have enriched the dominant species in the intestinal flora, with α-diversity analysis revealing similar results. Chao1 and Ace diversity indices indicated that intake of α-CD and β-CD reduced the richness of gut microbes, whereas γ-CD exerted the opposite effects. Shannon and Simpson indices reflect differences in community diversity for each group of samples. The intake of α-CD and γ-CD partially restored communal diversity, which had been destroyed by the high-fat diet without dietary cellulose. Moreover, β-CD ingestion reduced communal diversity, implying that the dominant species may have been enriched further. β-diversity analysis showed similarities in microbial composition between groups. High-fat diets without dietary cellulose significantly affect fecal microbial diversity. Although cellulose supplementation can restore microbiota to levels of a low-fat diet, supplementation with α-CD and β-CD has been associated with marked alteration in gut microbiota.

Species composition of intestinal flora for each group of mice was further analyzed. At the phylum level, we focused on the relative abundance of Firmicutes and Bacteroidetes, which are the primary bacteria in metabolism of undigested food residues and help in the digestion of dietary fibers and polyphenols by a complex metabolic energy-harvesting mechanism [50]. Differences between the fiber group and ACD, BCD, and YCD groups were mainly in the relative abundance of Firmicutes, which may be one of the reasons for the different effects of cellulose and CDs on fat accumulation. The F/B ratio is an index of the health of gut microbes, which bring about an increase in Firmicutes abundance and a decrease in Bacteroidetes abundance in obese mice [30]. The high-fat diet without cellulose significantly increased the F/B ratio by upregulating the relative abundance of Firmicutes and downregulating that of Bacteroidetes, whereas the addition of cellulose and CDs brought the ratio back down to normal levels. At the genus level, the advantages of CDs were mainly associated with Lactobacillus and Akkermansia. Lactobacillus is involved in the alleviation of obesity via the modulation of lipid cholesterol metabolism and the involvement of microbiota metabolites, including SCFAs, LPS, and antimicrobial substances [51,52]. Akkermansia colonizes the host gut mucosa layer and adjusts basal metabolism; therefore, it is considered in the development of next-generation therapeutic agents of obesity [53,54]. In addition, differences in inhibition of white fat accumulation between α-CD and other CDs may be attributed to the low relative abundance of Bifidobacterium. Despite the differences in the effects of the three CDs on species composition of intestinal flora, they were clustered into one group based on the dominant flora. Although the intake of cellulose can restore the original dominant flora destroyed by a high-fat diet without dietary cellulose, CDs exerted different effects on dominant flora.

Due to the similarity in effects of the three CDs on dominant flora, it is important to establish biomarkers for the effects of CDs. Akkermansia was found to be a biomarker for the ACD group, whereas Coprococcus and Butyricicoccus were biomarkers for the YCD group. Coprococcus and Butyricicoccus are a class of butyrate-producing bacteria; their metabolites strengthen epithelial barrier functions by increasing transepithelial resistance, thereby reducing incidences of inflammatory bowel disease [55]. These biomarkers have shown that γ-CD has a more significant effect in regulating intestinal health. Moreover, α-CD was shown to slightly reduce amylolytic hydrolysis of γ-CD and linear dextrins [56]. These pieces of evidence indicate that a combination of multiple CDs may be able to promote the host’s intestinal health.

With regards to the prediction of metabolic functional pathways, enhancement of metabolism of terpenoids and polyketides, as well as transport and catabolism functions, may be associated with inhibition of fat accumulation under high-fat dietary conditions by CDs. Terpenoids possess antitumor, anti-inflammatory, antibacterial, antiviral, and antimalarial properties; moreover, they have been shown to promote percutaneous absorption, prevent and treat cardiovascular diseases, and control blood glucose levels [57]. A group of natural biomolecules known as polyketides have shown clinical benefits for their anticancer, antimicrobial, antioxidant, and anti-inflammatory properties [58]. Upregulated expressions of secondary metabolites due to CD supplementation indicates its beneficial effects on intestinal health. Enrichment of genes involved in glycan biosynthesis and metabolism, as well as carbohydrate metabolism, in the BCD and YCD groups may be associated with the beneficial effects of β-CD and γ-CD on lipid levels, lipid droplet accumulation in cells, and short-chain fatty acid production.

## 5. Conclusions

In summary, dietary CD supplementation is an effective strategy for obesity prevention. This dietary pattern exhibited better results than cellulose supplementation in non-obese mice on a high-fat diet. Three CDs supplements—α-CD, β-CD, and γ-CD—showed different efficacies in slowing fat accumulation, promoting energy expenditure, increasing SCFA levels, and altering microbiota composition, suggesting that a combination of multiple CDs may play an important role in prevention and management of obesity. Although β-CD has shown excellent advantages in preventing fat accumulation, the hepatotoxicity it causes is still not negligible. We only used serum AST and ALT levels to reflect hepatotoxicity, which is inadequate, and more analyses and tests should be used to assess the toxicity of β-CD on hepatocytes and hepatobiliary ducts. In addition, the effect of CDs on the expression of genes and proteins related to lipid metabolism should be further investigated to analyze their mechanism for the prevention of fat accumulation.

## Figures and Tables

**Figure 1 foods-11-01118-f001:**
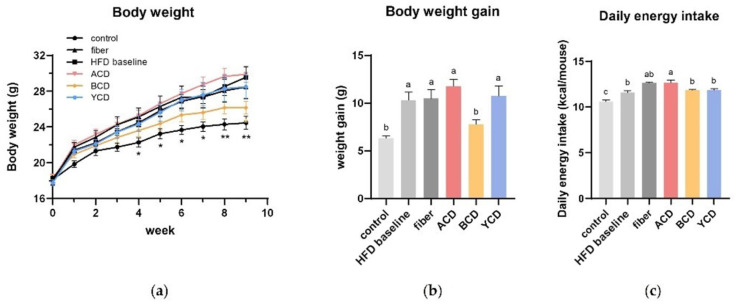
β-CD supplementation inhibited excess weight gain in C57BL/6J mice fed a high-fat diet independent of energy intake. (**a**) Changes in body weight from week 0 to week 9; (**b**) body weight gain at week 9 compared to week 0; (**c**) daily energy intake during the experimental period. n = 6. Control, low-fat diet; HFD baseline, high-fat diet without cellulose; fiber, high-fat diet; ACD, high-fat diet that replaced α-CD with cellulose; BCD, high-fat diet that replaced β-CD with cellulose; YCD, high-fat diet that replaced γ-CD with cellulose. Data are expressed as means ± SEM. * *p* < 0.05, ** *p* < 0.01 compared to the HFD baseline group (ANOVA). Values not sharing a common superscript differ significantly among groups (*p* < 0.05, ANOVA).

**Figure 2 foods-11-01118-f002:**
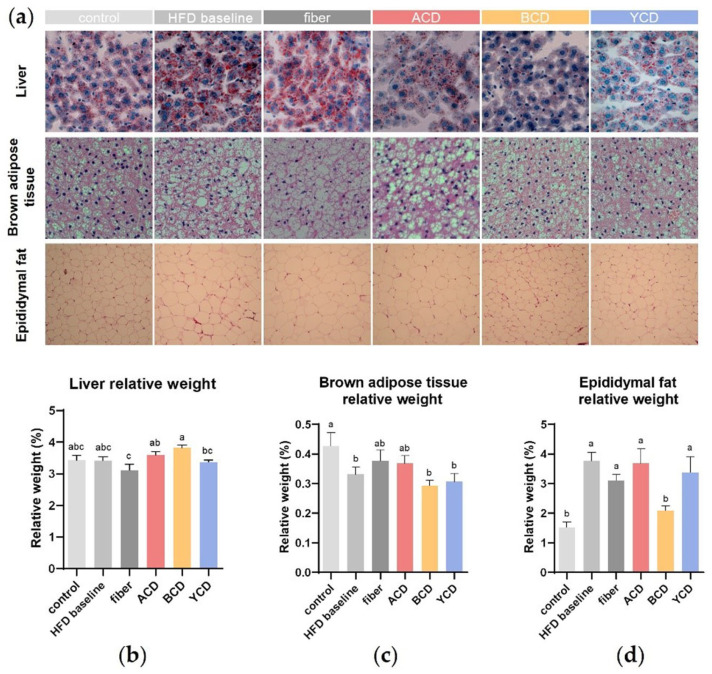
α-CD, β-CD, and γ-CD supplementation reduced fat accumulation in the liver, brown adipocytes, and epididymal adipose tissue. (**a**) Oil red staining of liver tissue sections and hematoxylin–eosin staining of brown adipose tissues and epididymal fat sections; (**b**) relative weights of the liver; (**c**) relative weights of brown adipose tissue; (**d**) relative weights of epididymal fat. n = 6. Control, low-fat diet; HFD baseline, high-fat diet without cellulose; fiber, high-fat diet; ACD, high-fat diet that replaced α-CD with cellulose; BCD, high-fat diet that replaced β-CD with cellulose; YCD, high-fat diet that replaced γ-CD with cellulose. Data are expressed as means ± SEM. Values not sharing a common superscript differ significantly among groups (*p* < 0.05, ANOVA).

**Figure 3 foods-11-01118-f003:**
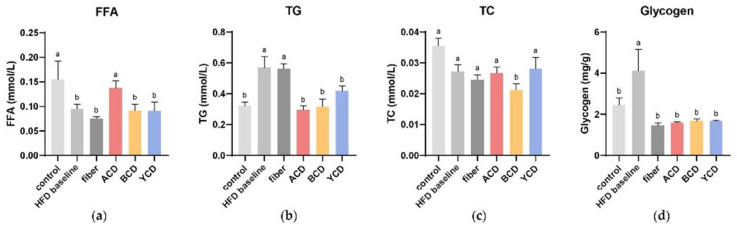
α-CD, β-CD, and γ-CD supplementation reduced liver glycogen and TG levels, and β-CD supplementation reduced TC levels in liver tissue. (**a**) FFA; (**b**) TG; (**c**) TC; and (**d**) glycogen content quantified in liver homogenate. n = 6. Control, low-fat diet; HFD baseline, high-fat diet without cellulose; fiber, high-fat diet; ACD, high-fat diet that replaced α-CD with cellulose; BCD, high-fat diet that replaced β-CD with cellulose; YCD, high-fat diet that replaced γ-CD with cellulose. Data are expressed as means ± SEM. Values not sharing a common superscript differ significantly among groups (*p* < 0.05, ANOVA).

**Figure 4 foods-11-01118-f004:**
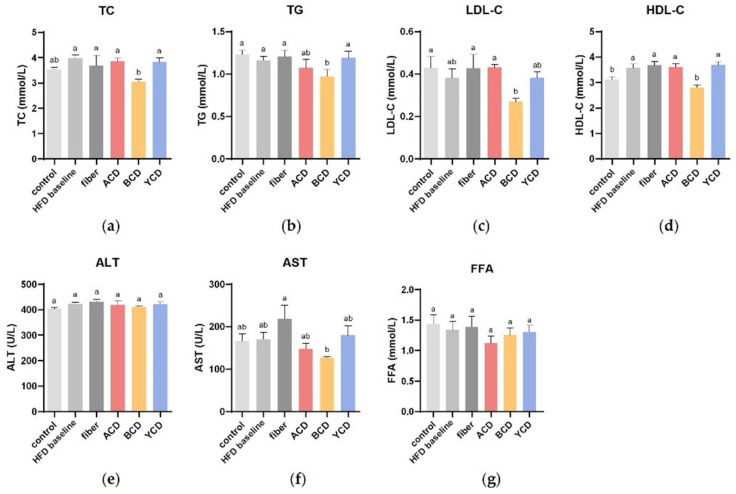
β-CD supplementation improved blood lipid parameters and did not cause damage to hepatocytes. Serum concentrations of (**a**) TC; (**b**) TG; (**c**) LDL-C; (**d**) HDL-C; and (**g**) FFA. Serum (**e**) ALT; (**f**) AST activities. n = 6. Control, low-fat diet; HFD baseline, high-fat diet without cellulose; fiber, high-fat diet; ACD, high-fat diet that replaced α-CD with cellulose; BCD, high-fat diet that replaced β-CD with cellulose; YCD, high-fat diet that replaced γ-CD with cellulose. Data are expressed as means ± SEM. Values not sharing a common superscript differ significantly among groups (*p* < 0.05, ANOVA).

**Figure 5 foods-11-01118-f005:**
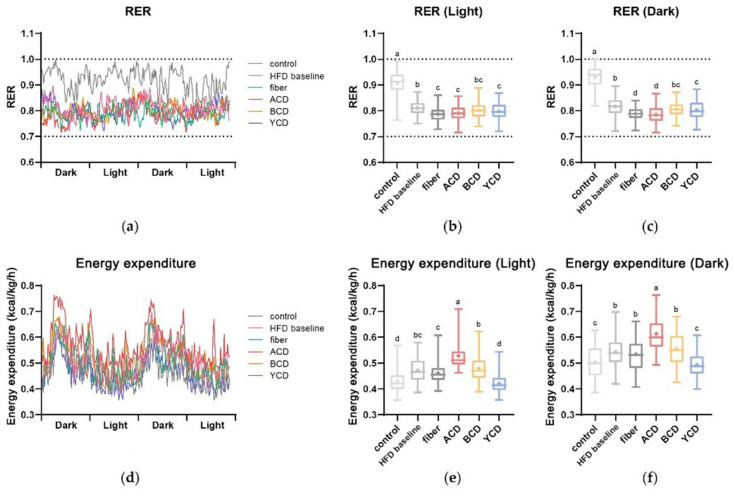
α-CD supplementation promoted energy expenditure, whereas α-CD and γ-CD improved the fat supply ratio. (**a**–**c**) Respiratory exchange ratio (RER) and (**d**–**f**) energy expenditure of C57BL/6J mice in 48 h. n = 3. Control, low-fat diet; HFD baseline, high-fat diet without cellulose; fiber, high-fat diet; ACD, high-fat diet that replaced α-CD with cellulose; BCD, high-fat diet that replaced β-CD with cellulose; YCD, high-fat diet that replaced γ-CD with cellulose. Data are expressed as means ± SEM. Values not sharing a common superscript differ significantly among groups (*p* < 0.05, ANOVA).

**Figure 6 foods-11-01118-f006:**
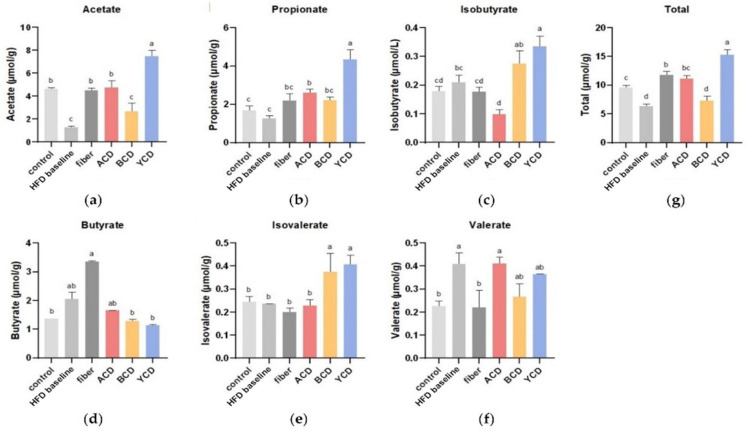
α-CD and γ-CD supplementation enhanced SCFA levels of cecum contents. Concentrations of (**a**) acetate; (**b**) propionate; (**c**) isobutyrate; (**d**) butyrate; (**e**) isovalerate; (**f**) valerate; and (**g**) total SCFAs in cecum contents. n = 6. Control, low-fat diet; HFD baseline, high-fat diet without cellulose; fiber, high-fat diet; ACD, high-fat diet that replaced α-CD with cellulose; BCD, high-fat diet that replaced β-CD with cellulose; YCD, high-fat diet that replaced γ-CD with cellulose. Data are expressed as means ± SEM. Values not sharing a common superscript differ significantly among groups (*p* < 0.05, ANOVA).

**Figure 7 foods-11-01118-f007:**
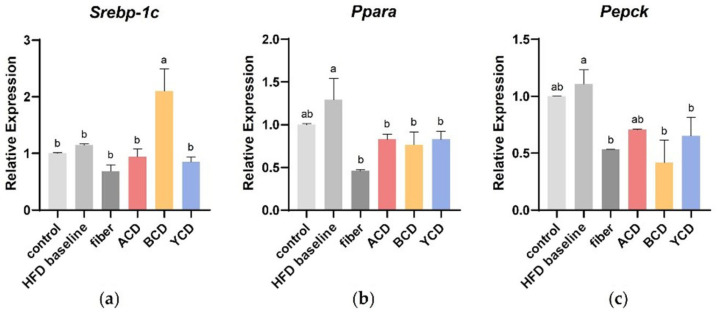
α-CD, β-CD, and γ-CD supplementation altered the expression of lipid metabolism-related genes. The relative expression of (**a**) *Srebp-1c* (sterol-regulatory element binding protein-1c); (**b**) *Pparα* (peroxisome proliferator activated receptor-α); and (**c**) *Pepck* (phosphoenolpyruvate carboxykinase) in liver tissue. n = 6. Control, low-fat diet; HFD baseline, high-fat diet without cellulose; fiber, high-fat diet; ACD, high-fat diet that replaced α-CD with cellulose; BCD, high-fat diet that replaced β-CD with cellulose; YCD, high-fat diet that replaced γ-CD with cellulose. Data are expressed as means ± SEM. Values not sharing a common superscript differ significantly among groups (*p* < 0.05, ANOVA).

**Figure 8 foods-11-01118-f008:**
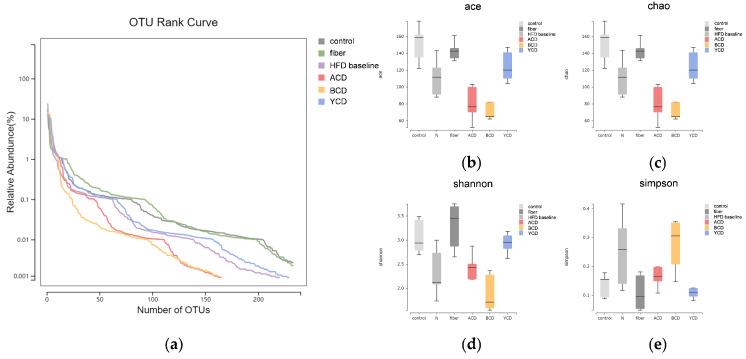
α-CD and β-CD supplementation reduced abundance and homogeneity within the gut microbiome. (**a**) OTU rank curve; (**b**) Ace index; (**c**) Chao index; (**d**) Shannon index; (**e**) Simpson index. n = 6. Control, low-fat diet; HFD baseline, high-fat diet without cellulose; fiber, high-fat diet; ACD, high-fat diet that replaced α-CD with cellulose; BCD, high-fat diet that replaced β-CD with cellulose; YCD, high-fat diet that replaced γ-CD with cellulose. The medians and interquartile ranges (IQRs) are shown in boxes. Whiskers indicate the lowest and highest values within 1.5 times the IQR of the first and third quartiles.

**Figure 9 foods-11-01118-f009:**
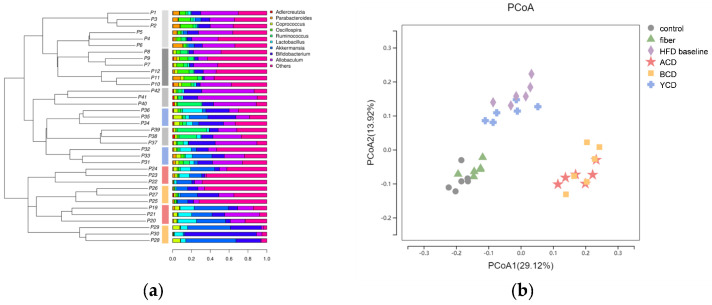
α-CD and β-CD supplementation resulted in inconsistent changes in gut microbial structure. (**a**) Unweighted pair-group method with arithmetic means cluster tree; (**b**) principal co-ordinate analysis, n = 6. Control, low-fat diet; HFD baseline, high-fat diet without cellulose; fiber, high-fat diet; ACD, high-fat diet that replaced α-CD with cellulose; BCD, high-fat diet that replaced β-CD with cellulose; YCD, high-fat diet that replaced γ-CD with cellulose.

**Figure 10 foods-11-01118-f010:**
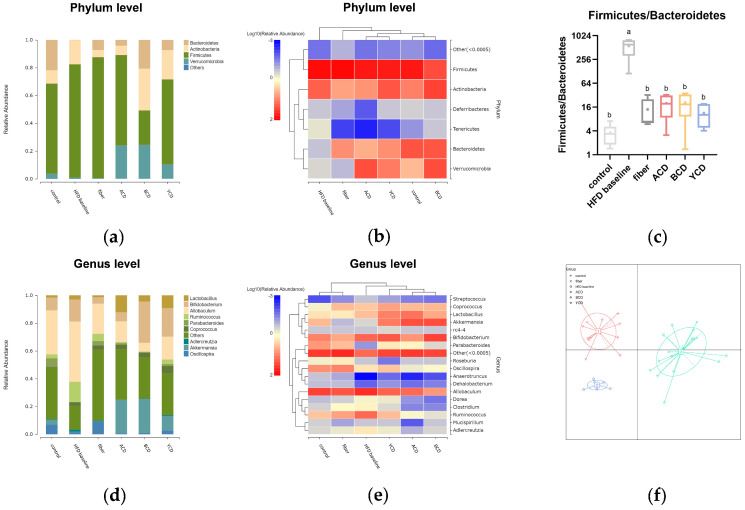
α-CD, β-CD, and γ-CD supplementation altered microbiota compositions. Fecal microbial community (**a**,**b**) at the phylum level and (**d**,**e**) at the genus level; (**c**) ratio of relative abundance of Firmicutes to Bacteroidetes; (**f**) enterotype clustering, n = 6. Control, low-fat diet; HFD baseline, high-fat diet without cellulose; fiber, high-fat diet; ACD, high-fat diet that replaced α-CD with cellulose; BCD, high-fat diet that replaced β-CD with cellulose; YCD, high-fat diet that replaced γ-CD with cellulose. Values not sharing a common superscript differ significantly among groups (*p* < 0.05, ANOVA).

**Figure 11 foods-11-01118-f011:**
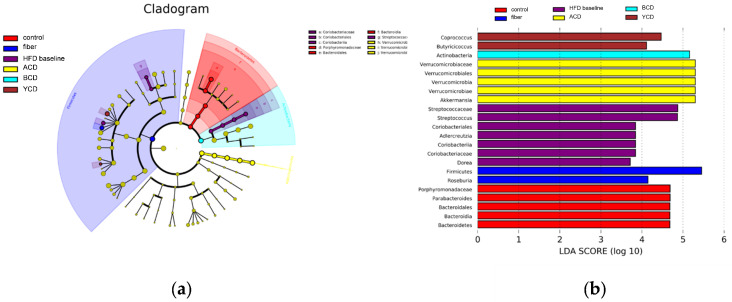
Akkermansia, Coprococcus, and Butyricicoccus were identified as biomarkers of CDs. (**a**) Linear discriminant analysis effect size (LEfSe) analysis; (**b**) linear discriminant analysis (LDA), n = 6. Control, low-fat diet; HFD baseline, high-fat diet without cellulose; fiber, high-fat diet; ACD, high-fat diet that replaced α-CD with cellulose; BCD, high-fat diet that replaced β-CD with cellulose; YCD, high-fat diet that replaced γ-CD with cellulose.

**Figure 12 foods-11-01118-f012:**
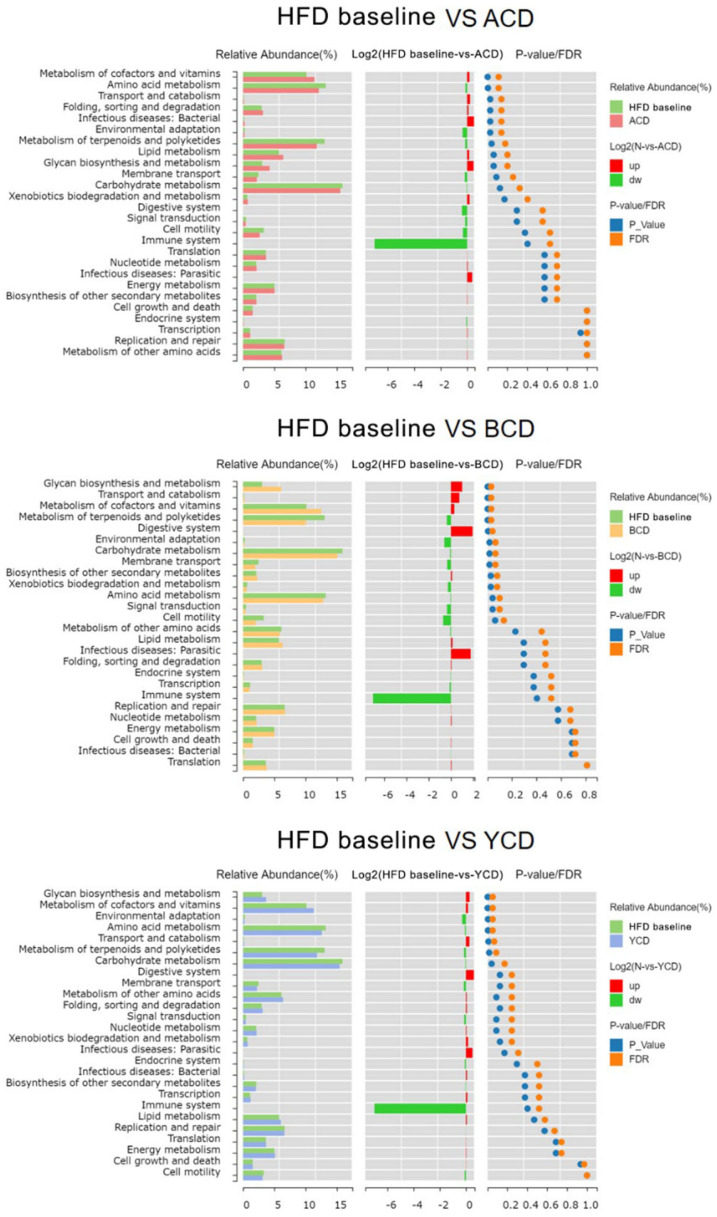
Effects of the three CDs on metabolism functional pathways. Control, low-fat diet; HFD baseline, high-fat diet without cellulose; fiber, high-fat diet; ACD, high-fat diet that replaced α-CD with cellulose; BCD, high-fat diet that replaced β-CD with cellulose; YCD, high-fat diet that replaced γ-CD with cellulose; n = 6.

**Table 1 foods-11-01118-t001:** Quantitative RT-PCR primer design.

Target Gene	Sequences (5′–3′)
*β* *-* *actin*	F: GGGTCAGAAGGACTCCTATG R: GTAACAATGCCATGTTCAAT
*Srebp-1c*	F: AACTTTTCCTTAACGTGGGCCT R: TGTCCAGTTCGCACATCTCG
*Ppar* *α*	F: ACTACGGAGTTCACGCATGTG R: TTGTCGTACACCAGCTTCAGC
*Pepck*	F: TCTTTGGTGGCCGTAGACCTG R: CCAGGTATTTGCCGAAGTTGTAG

## Data Availability

Data are contained within the article.

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
