# Peer review of "Beneficial Effects of Three Dietary Cyclodextrins on Preventing Fat Accumulation and Remodeling Gut Microbiota in Mice Fed a High-Fat Diet"

_foods, 2022, doi:10.3390/foods11081118_

Round 1

Reviewer 1 Report

The manuscript requires some editing. Suggestions are below:

  1. Title is vague- it sounds like the CD had beneficial effects on fat accumulation and not in its prevention- pls reword
  2. Intro line 48- unclear
  3. Procedure 2.2

: More info on the  experimental period- how long were the animals fed with the diet? and why?
how was feces collected to ensure it e was carried out under sterile conditions

: why 45 % fat- some literature suggest 60 %

: why fibre was used explain more about the fibre, type etc

: using the notation N- denotes that it is a normal diet- strongly suggest to change it to HFD
baseline data?

4. Results: is 9 weeks of supplementation sufficient?- 12 weeks recommended, pls discuss

  • Notations a,b,c on all graphs is  not explained Also mention sig difference on the graphs
  • lines 205-207- rephrase not clear
  • lines 220-222- rephrase not clear
  • lines 247-250: this is not good enough to establish hepatotoxicity- mention as limitation of study
  • line 263- RER in full
  • Fig 5d-f what is HEAT, explain
  • Interchange of use of fiber and cellulose- be consistent
  • limes 311-312 and Fig 7 legend and graph- ensure correct abbreviation of the genes
  • Include  a section in Discussion/ Conclusion , on the limitations and future work of this/from this study

Reviewer 2 Report

This is an interesting project and a well-prepared report.

Abstract Lines 19-21: Please include data on alpha-CD responses.

Abstract: Please note in Abstract if the responses to the three cyclodextrins are different, otherwise the last sentence is not logical. 

Lines 74-78: Please include a hypothesis for this study.

What do these doses of cyclodextrins represent in a human dose using the Reagan-Shaw equations? Is this a realistic dose in humans? 

Line 175: 20% is not a factor.
